# OpenReview forum: "Reliability-Aware Preference Learning for LLM Reward Models"
_TMLR — Rejected by TMLR_

### Review · Reviewer_Cq5M · 2025-11-10

**Summary Of Contributions:**

This paper addresses problem in aligning language models with human values i.e. unreliable human feedback and proposes reliability aware preference learning. modifies the standard loss function to incorporate Annotator Reliability Metrics (ARMs) for each comparison pair. This allows the model to amplify signals from reliable judgments and attenuate those from unreliable ones. The authors introduce two datasets: LIE (Length-Incentivized Evaluations) to elicit and study unreliable feedback , and TRUE (Testing Reasoning and Understanding Errors) to calibrate and generate the ARMs.

Strengths:
- The paper tackles a well known problem in RLHF: the unreliability of human annotators and its impact on the training

- Novel datasets (LIE, TRUE) providing controlled environment and principled way to calibrate the relibility metrics

Weakness:
- The paper's primary focus is on the trade-off between length and factual correctness. While this is a clear and important example, unreliability in human feedback is much broader (for example politeness vs honesty, complex reasoning). Not clear how well the findings can be generalize to more general and complex datasets.

- The TRUE dataset relies on relies on the existence of "objectively correct" answers this works well for factual questions (like those in LIE and TRUE) but is much harder for the alignment problems that RLHF is often used for, such as preferences about creativity, writing styles etc. The solution is tied to tasks where a GT for correctness exists.

- Scalability? the proposed solution requires new diverse dataset like TRUE for calibration to match the target domain. Is this more practical than other methods like AI-assisted evaluation?

**Audience:**

Yes

**Audience Explanation:**

Yes. Unreliable human feedback is a crucial problem need to tackle to improve the training

**Claims And Evidence:**

Yes

**Claims Explanation:**

The paper's first claim—that standard reward models (RMs) learn to overvalue superficial features like length over factual accuracy—is supported by experiments on their custom LIE dataset. They provide clear data showing that human annotators preferred longer, incorrect answers over 70% of the time and that a standard RM trained on this data inherited this bias, weighting length about 6 times more than correctness. The paper's second claim—that their proposed RAPL method can correct this bias—is supported by a second experiment. They use their TRUE dataset to create Annotator Reliability Metrics (ARMs) and show that when RAPL is applied to the same biased LIE data, the resulting RMs successfully prioritize factual correctness over length, as shown in Table 2.

**Requested Changes:**

How could this framework apply to preferences that lack a simple "ground truth"?

The paper should be more upfront about the cost of creating a TRUE-like dataset. A discussion on "cheaper" ways to generate ARMs would be valuable

The paper mentions that the framework could "potentially be applicable to direct preference optimization (DPO)". This is a strong claim that is left unexplored.

---

> ### Author Response · Authors · 2025-12-16
> **Response to Reviewer Cq5M**
>
> Thank you for your constructive feedback on our paper. We appreciate that you found our work valuable and are happy to address each of your questions below.
>
> **How could this framework apply to preferences that lack a simple "ground truth"?**
> To clarify, we don't actually apply RAPL to / train reward models on the TRUE dataset, which is the dataset with "ground truth" labels; these labels are used only to train the ARM predictor. Once trained, this ARM predictor can generate reliability estimates for any preference comparison, including those without objectively correct answers, based on features of the prompt and responses that correlate with annotator reliability (e.g., cognitive difficulty, domain complexity, susceptibility to bias). In other words, TRUE only needs to be constructed once. As long as the TRUE dataset annotations are collected in a similar fashion to the target preference learning dataset—using a similar format, providing similar instructions to annotators, etc.—the resulting ARM model will learn patterns about when annotators struggle that transfer to the new domain.
>
> We also note that RAPL does not actually require that final preferences have ground truth. In fact, our experimental setting with the LIE dataset involves both "objective" elements (factual correctness) and "subjective" elements (preferences over length/detail). We chose this controlled setting specifically so we could rigorously evaluate whether RAPL successfully recovers annotators' true underlying preferences for factually correct answers despite their expressed preferences being unreliable.
>
> **The paper should be more upfront about the cost of creating a TRUE-like dataset. A discussion on "cheaper" ways to generate ARMs would be valuable**
> This is certainly a valid practical concern to raise; however, we want to emphasize two points:
>
> * First, as noted above, TRUE is a one-time investment. The calibration dataset does not need to be recreated for each new preference learning task, so long as the annotation-collection protocol is consistent with that of the target preference learning dataset. Given that we follow standard data collection procedures, practitioners can likely leverage our existing TRUE-trained models or simply extend our dataset incrementally.
> * Second, our results demonstrate that even simple behavioral indicators like annotator-reported confidence, when calibrated using the TRUE dataset, yield meaningful improvements (Table 2, "Calibrated Confidence" rows). This suggests a practical middle ground: practitioners could use relatively lightweight behavioral signals calibrated with TRUE data, rather than training dedicated ARM prediction models.
>
> **The paper mentions that the framework could "potentially be applicable to direct preference optimization (DPO)". This is a strong claim that is left unexplored.**
> You are correct that this claim was insufficiently supported. We have removed this statement from the paper. While we believe there is a principled way to extend RAPL to DPO (since DPO implicitly optimizes a similar preference likelihood), validating this claim is beyond the scope of the current work. We leave this as a direction for future research.

---

### Review · Reviewer_tKQM · 2025-11-15

**Summary Of Contributions:**

The paper introduces Reliability-Aware Preference Learning (RAPL), a framework for training reward models that accounts for the fact that human annotators vary widely in their reliability when comparing model responses. To study how unreliable feedback distorts reward learning, the authors create LIE, a controlled dataset where response length and correctness are decoupled, and show that standard preference learning mistakenly prioritizes superficial cues (like longer answers) over factual accuracy when annotators struggle. They also introduce TRUE, a calibration set used to estimate per-example annotator reliability through confidence, response time, or an LLM predictor. RAPL integrates these reliability scores directly into the preference-learning loss—either by scaling preference strength or by modelling annotators as partially random —which down-weights low-reliability comparisons. Experiments demonstrate that RAPL yields reward models that emphasize correctness rather than length and achieve better alignment on noisy human preference data.

**Audience:**

Yes

**Audience Explanation:**

The findings of this paper would be of strong interest to the TMLR audience, particularly researchers working on RLHF, reward modeling, dataset curation, and human–LLM interaction. The paper addresses a foundational and increasingly relevant issue in preference learning—the variability of annotator reliability—and provides both conceptual insight and practical methodology for mitigating its negative effects. As reward models play a central role in aligning modern LLMs, understanding how unreliable human judgments distort learned preferences is valuable to both practitioners and theorists. The introduction of the LIE and TRUE datasets, along with the RAPL framework, offers tools and empirical evidence that many TMLR readers could directly build upon in their own work.

**Claims And Evidence:**

Yes

**Claims Explanation:**

The submission’s claims are supported by accurate, convincing, and clearly presented evidence. The authors motivate the problem of unreliable human preference annotations with empirical observations, then introduce two carefully designed datasets—LIE and TRUE—that explicitly isolate annotator reliability and allow controlled evaluation. The experimental analyses consistently demonstrate that standard preference learning overfits to superficial features (e.g., response length) when annotator judgments are unreliable, while the proposed RAPL framework systematically reduces this effect across multiple reliability estimators. The evidence is thorough: the paper includes ablation studies, comparisons across different ARM estimators (confidence, response time, LLM-based predictors), and evaluations on both synthetic controls and external benchmarks, all of which converge to support the central claim that reliability-aware weighting leads to more robust reward models. Overall, the empirical results align well with the theoretical framing, and the evidence is clearly and convincingly presented.

**Requested Changes:**

1.  The paper could include qualitative examples or typologies of “unreliable” comparisons, which would help readers understand what kinds of errors RAPL mitigates.

2. Since ARM estimation adds cost, a brief runtime/memory analysis would be helpful for practitioners.

3. Is there any specific real-world setting that the method proposed in the paper could be applied to? For example, is there a clear threshold or criterion that determines when RAPL should be used instead of the standard Bradley–Terry approach?

---

> ### Author Response · Authors · 2025-12-16
> **Response to Reviewer tKQM**
>
> Thank you for your thorough review of our work. We address each of your requested changes below.
>
> **The paper could include qualitative examples or typologies of “unreliable” comparisons, which would help readers understand what kinds of errors RAPL mitigates.**
> We address this point in a couple of places within our paper:
> * In the second paragraph of Section 1, we list different failure modes of RLHF-trained LLMs that stem from unreliable preference data. These include: hallucinating factually inaccurate outputs, producing sycophantic text that simply agrees with whatever the user says, and imitating persuasion and manipulation tactics, sometimes attempting to convince humans they are correct when they aren't. We explain how these issues arise because learned reward functions tend to prioritize superficial metrics like length over important factors like accuracy.
> * Table 1 provides concrete examples from our LIE dataset. The table shows that annotators are much more likely to pick the incorrect answer when it is associated with a stylistic feature that they prefer--in this case, length. For instance, when comparing short and correct responses against long and incorrect responses, annotators chose the long and incorrect option 354 out of 542 times.
>
> **Since ARM estimation adds cost, a brief runtime/memory analysis would be helpful for practitioners.**
> We note that the models trained on our TRUE dataset do not need to be re-trained for every target preference learning dataset, so long as the annotation protocol is consistent. Additionally, for estimating ARMs with the TRUE model, we need only one forward pass per preference pair. This is less than half the cost of one epoch of reward model training, which requires two forward passes (one for the chosen and one for the rejected response) plus a backward pass for each preference pair.
>
> **Is there any specific real-world setting that the method proposed in the paper could be applied to? For example, is there a clear threshold or criterion that determines when RAPL should be used instead of the standard Bradley–Terry approach?**
> RAPL is applicable to any setting where the failure modes of RLHF-trained models that stem from unreliable annotator feedback are a concern, such as sycophancy, length bias, and hallucination. Several past works cited in our paper note that annotators are unreliable in these contexts, and through our own LIE dataset collection, we were also able to validate this fact (Table 1).
>
> As for a threshold, besides the slight increase in computation time, there is no significant downside to using RAPL even if comparisons are already reliable--in such cases, the ARMs will simply be consistently high, and RAPL will behave similarly to standard preference learning.

---

### Review · Reviewer_aHip · 2025-12-01

**Summary Of Contributions:**

This paper studies the reliability issue in the reward model learning process in RLHF. In particular, the paper first constructed the LIE dataset with negative correlation between response length and correctness to show that annotator tend to prefer longer answers, despite of the correctness. The paper then proposed the Reliability-Aware Preference Learning by modifying the objective based on Bradley-Terry model to incorporate the annotator reliability metric. To learn the annotator reliability metric, the paper proposed method to fit a logistic regressor. The paper experimentally showed that the proposed method performs better than standard preference optimization and preference optimization that incorporates raw confidence score on the LIE dataset.

**Audience:**

Yes

**Audience Explanation:**

RLHF is an important approach for LLM fine-tuning, and the issue of response reliability is indeed a significant problem.

**Broader Impact Concerns:**

None.

**Claims And Evidence:**

No

**Claims Explanation:**

There are several key points I found confusing in this paper:

1. First, the paper state their contribution as "We find that reward models trained on unreliable human feedback tend to place higher weight on obvious proxies like length and less weight on factual correctness". However, instead of investigating a wide range of proxies, the paper only studied the influence of length.

2. It is not clear whether the dataset TRUE and LIE are biased by the same set of annotator. If so, the ARM regressor learned on TRUE will naturally perform well on LIE, but intuitively can hardly transfer to unseen annotators.

3. From Table 2 it is hard to interpret why $\beta$-RAPL performs better than standard preference learning. Also, it is unclear how to interpret the negative values in the metric.

4. It is unclear what kind of features are collected and used as input in the ARM regressor.

5. Experiments needs to be conducted on other datasets to show the generality of the method.

**Requested Changes:**

1. I suggest clarifying the contribution by only mentioning the impact of response length instead of broadly stating cognitive bias or obvious proxies.

2. The paper should clarify the method for choosing the annotators of the two datasets. The paper should also clarify on what features are used in the ARM score regressor.

3. The paper should comment on how to interpret the negative values of the metric in Table 2.

4. The paper should consider adding more experimental results on different datasets.

---

> ### Author Response · Authors · 2025-12-16
> **Response to Reviewer aHip**
>
> Thank you for your review of our paper. We appreciate the fact that you find our work of importance, and we address your questions below:
>
> **The paper only studied the influence of length and is limited in its experimentation.**
> We also present RewardBench results in Table 3. Even though the LIE training dataset emphasizes length as the main cognitive bias, RewardBench is a well-established benchmark that evaluates reward models across multiple categories: chat, safety, and reasoning. RAPL-trained models show improvements across multiple RewardBench categories. This suggests that RAPL helps not just in the length bias case: annotators are likely unreliable in other ways as well, and RAPL improves preference learning more broadly.
>
> Furthermore, we expect RAPL to generalize to other biases because the framework itself is agnostic to the specific source of unreliability--it simply down-weights comparisons where annotators are unlikely to provide accurate feedback, regardless of the underlying cause (cognitive biases, knowledge gaps, survey design, etc.).
>
> **It is unclear whether TRUE and LIE are biased by the same set of annotators, and whether ARMs can transfer to unseen annotators.**
> TRUE and LIE were collected using the same annotation protocol (same format, similar instructions, same platform), but not necessarily the same individual annotators. The key requirement for transfer is consistency in the annotation-collection protocol. Since we follow standard data collection procedures (as described in Appendix A.2 and B.1), practitioners can likely leverage our existing TRUE-trained models directly or extend our dataset incrementally.
>
> If practitioners have a substantially different annotator population, they can collect new annotations using our existing TRUE dataset prompts and retrain the ARM models. The TRUE dataset itself is reusable--only the ARM model may need to be retrained to calibrate to new annotators.
>
> **It is hard to interpret why β-RAPL performs better than standard preference learning, and how to interpret the negative values in our results.**
> We want to clarify that β-RAPL does not consistently perform better than standard preference learning, and that this is actually one of our key findings. We explored two variants of RAPL and found that only ε-RAPL reliably improves over standard preference learning. We have adjusted the language in our contributions at the end of the introduction to emphasize that we explore two alternatives and find that only one handles unreliable annotations effectively.
>
> The reason β-RAPL struggles is explained in Section 6.2: β-RAPL scales the gradient magnitude but does not change its direction. This means unreliable comparisons still influence the optimization, just to a lesser degree. In contrast, ε-RAPL can actually drive gradients toward zero for low-reliability comparisons, effectively silencing their signal. Figure 3 illustrates this with a toy example showing how ε-RAPL moves the loss minimum much further in the correct direction than β-RAPL.
>
> The negative weight values in Table 2 (e.g., β-RAPL with raw confidence) indicate the model is placing slightly negative weight on correctness. These values are very close to zero and likely reflect instability when using uncalibrated ARMs—raw confidence scores are poor predictors of annotator reliability, as annotators tend to be overconfident even when making incorrect choices (as noted in Section 6). This is precisely why calibration using the TRUE dataset is important.
>
> **It is unclear what kind of features are collected and used as input in the ARM regressor.**
> Sorry for any confusion here. For defining ARMs, we use two approaches:
> * **Fine-tuned LLMs**: We fine-tune Llama 3 8B on the TRUE dataset using a binary classification loss. The full prompt-response pairs are used as input, and the model predicts whether an annotator will make the correct choice.
> * **Calibrated Behavioral Indicators**: For annotator confidence and decision time, we fit a simple one-variable logistic regression on the TRUE dataset that maps the indicator value to the probability of annotator correctness.

---

### Decision · Action_Editor_SDLT · 2026-01-10

**Recommendation:** Reject

**Additional Comments:**

1. Out-of-LIE validation is the key missing evidence loop for the current method framing; demonstrating consistent results on real preference datasets under a realistic RM training setup would significantly strengthen the contribution and clarify whether the observed gains extend beyond the LIE setting.

2. Another promising direction is to position ARMs as a practical way to prioritize or filter high-value preference pairs.

3. It would also be valuable to more clearly disentangle feature effects (e.g., length vs. correctness) and alternative explanations for the observed improvements, to better characterize the mechanism of the method.

**Audience:**

Yes

**Audience Explanation:**

Many individuals in TMLR’s audience would be interested in this work. Unreliable preference labels are a central issue in RLHF and reward modeling, and the proposed framing of per-comparison reliability (ARMs), along with the LIE/TRUE diagnostic setup, is relevant to researchers working on preference learning, evaluation, and data curation.

At the same time, given the current evidence is primarily based on the LIE setting with limited out-of-distribution validation, the findings are best interpreted as diagnostic insights rather than fully established method-level conclusions.

**Claims And Evidence:**

No

**Claims Explanation:**

While the topic is significant, the submission currently does not meet TMLR's standards for Technical Correctness and Claims supported by Evidence:

Generalization Gap: The method is validated only on the internal LIE dataset. Without out-of-distribution (OOD) experiments, it is unclear whether the method generalizes beyond the specific biases of the LIE construction. During discussion, the authors clarified that they would not expect clean transfer to OOD tasks, which limits the empirical support primarily to the LIE setting.

Length-Proxy Risk: Analysis of the reported results suggests that the observed performance gains are strongly associated with increased reliance on response length. This introduces a plausible alternative explanation in which improvements may arise from coupling with length rather than from improved robustness. Based on the discussion, the current evidence does not clearly distinguish between these possibilities.

Interpretability: Key metrics in Table 2 remain difficult to interpret in isolation. The current evidence is not sufficient to clearly separate genuine robustness improvements from potential configuration sensitivity or shortcut learning effects.

Conclusion: The claims are not supported by clear and convincing evidence under the current experimental setup. Strengthening the paper would require validation beyond LIE and clearer empirical disentanglement of feature effects (e.g., length vs. correctness).

**Resubmission Of Major Revision:**

The authors may consider submitting a major revision at a later time.